# Power Consumption Forecast of Three Major Industries in China Based on Fractional Grey Model

**Yuhan Xie [1], Yunfei Yang [1,2] and Lifeng Wu [1,*]**

1  School of Management Engineering and Business, Hebei University of Engineering, Handan 056038, China
2  School of Engineering Science, University of Chinese Academy of Sciences, Beijing 100049, China
*  Correspondence: wulifeng@hebeu.edu.cn

**Abstract:** As one of the most significant carbon emission departments in China, the power industry will gradually become the core hub of reducing carbon emissions in the process of undertaking carbon emissions transferred from other industries. Therefore, it is of vital importance to predict the power consumption in China's end energy consumption to achieve the carbon peak goal on time. This paper firstly uses the gray relational analysis model to study the relationship between power consumption indicators of the three major industries and some social and economic indicators and obtains the influencing factors with the greatest correlation with the power consumption of the three industries. Then, based on the analysis of socio-economic factors, considering different growth rates, the GMCN(1,N) model of electricity consumption in China's three major industries is established. Forecast data under different scenarios have important practical significance for formulating active and effective energy policies. The data indicate that the secondary and tertiary industries consume the greatest amount of electricity. It is estimated that the power consumption of China's three major industries will reach 10.15 trillion kWh (kilowatt hours) by 2030.

**Keywords:** three major industries; power consumption; gray correlation analysis model; GMCN(1,N) model

## 1. Introduction

As the foundation and guarantee of social development, energy has very important economic and social value [1]. With the rapid economic and social development, the energy problem has become increasingly prominent, and the energy consumption and its future development trends have attracted extensive attention from scholars at home and abroad. Xue Yuexin et al. predicted Beijing's total energy consumption will reach 85.728 million tons in 2025, providing reference and guidance for Beijing's energy development planning [2]. Sun Manli et al. predicted that in 2025, the total installed capacity of natural gas distributed energy in China will reach 33.655 million kilowatts, 36.385 million kilowatts, and 40.25 million kilowatts, respectively, under high, medium, and low scenarios [3]. Jiao Jianling predicted China's future oil demand in the context of carbon emission reduction [4]. Wang Bing et al. estimated that China's coal consumption will reach 560 million to 760 million tce in 2060, which will help guide the development trend of coal energy [5].

With the continuous acceleration of urbanization and industrialization in China, the rapid economic and social development has increased the demand for energy, especially electric energy. Electricity will be the mainstay of energy growth in the next 10 years [6]. Al-Musaylh et al. proved that the MARS and SVR models are more suitable for the Queensland electricity demand study than the ARIMA model [7]. Yu Xiaoxiao used gray forecasting and time trend extrapolation to longitudinally analyze the electricity consumption data in Anhui Province, and made short-term forecasts for electricity demand [8]. Wu et al.

established a new multivariate grey prediction model considering the total population to predict the electricity consumption in Shandong Province [9]. Ren Fangling et al. selected the annual electricity consumption in Shaanxi Province as the research object and achieved good prediction results through the optimized multiple linear regression model and gray GM model [10]. Nie Jing used the LMDI model to explore the power consumption intensity of Beijing–Tianjin–Hebei [11]. Su Zhenyu established a multiple linear regression equation to forecast the electricity demand in Suzhou [12]. Li Wencong used the optimized GM(1,1) model to explore and analyze the electricity consumption of the whole society in Hubei Province [13]. Chen Tingting analyzed the terminal energy consumption in Shaanxi Province by improving the variable weight combination forecasting model and provided suggestions for the development of the electricity market in Shaanxi Province [14]. Zhai Ying et al. analyzed the relationship between the cyclical fluctuation of per capita electricity consumption and various influencing factors in various regions of China [15]. Williams et al. performed a piecewise interpolation operation on the time series of discrete observations, developed a power demand forecasting method, and provided a reliable forecasting method for short-term power consumption [16]. Lebotsa proposed a mixed model of bounded variables to predict short-term electricity consumption [17]. Wang Yanbo proposed a comprehensive forecasting method for monthly electricity consumption based on the STL decomposition model [18]. Hongye Guo et al. proposed a monthly electricity consumption prediction framework based on a vector correction model and explored the potential connection of energy footprint [19]. Based on grey forecasting and time series methods, respectively, Xiao Zheng et al. constructed grey Verhulst–ARIMA forecasting and seasonal ARIMA models to predict seasonal electricity consumption [20]. Gao Hong et al. used the GM(1,1) model to predict and analyze the power consumption of China's manufacturing industry and put forward corresponding suggestions [21]. Ma Chao et al. took Tianjin's manufacturing industry as the main body and made predictions after analyzing the development trend of electricity consumption, which provided a reference for the construction of the Tianjin power grid [22]. Through the scenario analysis method, Liu Ruiyu et al. researched the medium and long-term development potential of electricity consumption in China's high-energy-consuming industries under seven scenarios, indicating that high-energy-consuming industries still have a great pulling effect on the economy [23]. Qu Bo et al. analyzed the demand of energy and electricity consumption side under the background of "carbon neutrality", explored the development direction of energy green consumption transformation, and probed into new development models of the electricity consumption side from different fields such as industry, construction, transportation, and agriculture [24].

The gray correlation model analyzes the development trend of certain factors and determines the correlation sequence according to the correlation degree and the correlation matrix. The closer the two curves are, the greater the grey correlation degree between the corresponding sequences is, and vice versa. Compared with traditional mathematical statistical methods, the grey relational analysis model has fewer observations, and does not require a large sample size, nor does it require typical distribution laws; the calculation is simple, and the quantitative analysis results will be consistent with the qualitative analysis results. The gray prediction model is mainly suitable for solving problems with a small amount of data [25]. Compared with the univariate model, the grey multivariate convolution model with new information priority takes into account the influence of other variables on the development trend of certain factors, which further improves the accuracy and correlation of prediction. The GMCN(1,N) model gives greater weight to the new information in the calculation process, making full use of the value of the new information and making future predictions more accurate.

Based on the above research results, many scholars have made short-term forecasts of energy and power from the overall perspective of provinces, cities, and regions, while the classification forecast and medium and long-term development trend of electricity consumption in the three major industries have been less involved. The reason why the

forecast is divided into three parts is mainly because different industries are involved in different sectors and industries in the huge national economic system, and the influencing factors of power consumption are also diverse. Therefore, this paper divides China's production electricity consumption into three parts, namely, the electricity consumption of the primary industry, the electricity consumption of the secondary industry, and the electricity consumption of the tertiary industry, to forecast separately. In the past, autoregressive models and support vector models were used to solve such problems, but these statistical models have high requirements in terms of the amount of data and data distribution and are not suitable for samples with small data. Hence, many scholars use the gray model to solve the problem of small sample size. This paper also used an improved grey multi-variable prediction model to predict electricity consumption in China's three major industries. The main contributions of this paper can be summarized into three parts: (1) A grey multivariate convolution model is used with less error and priority of new information (GMCN(1,N)) to predict the power consumption of China's three major industries. (2) In order to accurately predict the power consumption of the three major industries, seven factors closely related to power consumption are selected for grey correlation analysis, and the key indicators affecting the power consumption of each industry are screened out. Then, the impact indicators are forecasted by scenario. (3) By forecasting the power consumption of the three major industries, it objectively reflects the energy and power consumption trends of the three major industries in China at different levels of development. The research results have important practical significance for the country to explore the real relationship between economic and social development and power consumption, plan the smart grid, and formulate active and effective energy policies.

The other main contents of this paper are as follows: The data sources are introduced in Section 2. The research methods and principles are given in Section 3, and Section 4 predicts the electricity consumption of China's tertiary industry under different per capita GDP growth rates. The power consumption of China's secondary industry under different urbanization levels is given in Section 5, and Section 6 studies the power consumption of the primary industry. The conclusions and prospects are shown at the end of the paper.

## 2. Data Source

The data of power consumption of China's primary, secondary, and tertiary industries are from the China Power statistical yearbook, and the output value of China's three major industries' economic and social data are all from China's statistical yearbook. We put China's economic and social data from 2011 to 2020 on github, and the usage data can be downloaded through the link https://github.com/han616807/hello (accessed on 26 July 2022).

The power consumption of China's three major industries from 2011 to 2020 is shown in Table 1, and the economic and social data of China from 2011 to 2020 is shown in Table 2.

**Table 1.** China's industrial electricity usage from 2011 to 2020.

| Indicators Year | 2011 | 2012 | 2013 | 2014 | 2015 | 2016 | 2017 | 2018 | 2019 | 2020 |
|---|---|---|---|---|---|---|---|---|---|---|
| The primary industry (billion kWh) | 1014 | 1003 | 1027 | 1014 | 1040 | 1093 | 1175 | 747 | 779 | 859 |
| The secondary industry (billion kWh) | 35,382 | 36,733 | 39,332 | 41,770 | 41,442 | 42,567 | 44,922 | 48,123 | 49,595 | 51,318 |
| The tertiary industry (billion kWh) | 5105 | 5693 | 6275 | 6671 | 7166 | 7973 | 8825 | 10,839 | 11,861 | 12,091 |

**Table 2.** China's economic and social data from 2011 to 2020.

| Year | Per Capita GDP (Yuan) | Total Population of the Country (10,000 People) | Urbanization Level (%) | Proportion of Primary Industrial Structure (%) | Proportion of Secondary Industrial Structure (%) | Proportion of Tertiary Industrial Structure (%) | Total Carbon Emissions (Billion Tons) |
|---|---|---|---|---|---|---|---|
| 2011 | 36,277 | 134,916 | 51.83 | 9.2 | 46.5 | 44.3 | 88.23 |
| 2012 | 39,771 | 135,922 | 53.10 | 9.1 | 45.4 | 45.5 | 90.00 |
| 2013 | 43,497 | 136,726 | 54.49 | 8.9 | 44.2 | 46.9 | 92.43 |
| 2014 | 46,912 | 137,646 | 55.75 | 8.6 | 43.1 | 48.3 | 92.88 |
| 2015 | 49,922 | 138,326 | 57.33 | 8.4 | 40.8 | 50.8 | 92.75 |
| 2016 | 53,783 | 139,232 | 58.84 | 8.1 | 39.6 | 52.4 | 92.74 |
| 2017 | 59,592 | 140,011 | 60.24 | 7.5 | 39.9 | 52.7 | 94.63 |
| 2018 | 65,534 | 140,541 | 61.50 | 7.0 | 39.7 | 53.3 | 96.49 |
| 2019 | 70,078 | 141,008 | 62.71 | 7.1 | 38.6 | 54.3 | 98.06 |
| 2020 | 72,000 | 141,212 | 63.89 | 7.7 | 37.8 | 54.5 | 98.94 |

## 3. Methods and Principles

### 3.1. Grey Correlation Analysis Model

The grey correlation degree model determines the correlation order according to the correlation degree and the correlation matrix. The closer the two curves are, the greater the grey correlation degree between the corresponding sequences is, and vice versa. In the 1980s, the grey model proposed by Professor Deng Julong was used to overcome the uncertainty issues caused by limited sample sizes and inadequate data [26] and was utilized in various fields. For example, Zhang Liyuan et al. studied the relationship between the level of rural revitalization and urban economic diffusion in eastern and central and western China by establishing a grey relational analysis model [27]. Wei Yuqi et al. used the grey relational analysis method to study the relationship between Dalian port logistics and regional economic development, which has a certain practical significance for improving Dalian's regional economic benefits [28]. Wang Xianqing et al. classified and weighted the main influencing factors of water resources security assessment and established a grey relational analysis model [29]. Since other factors such as population, economy, industry, and ecology affect the power consumption, these intricate factors can be understood as a grey system. Therefore, this paper uses the grey correlation analysis technique to study and analyze the power consumption of China's primary industry, secondary industry, and tertiary industry and China's social and economic indicators. To explore the factors that affect the changes of electricity consumption in China's three major industries, a grey correlation analysis model between China's three major industry electricity consumption indicators and seven social and economic indicators is established. The primary industry lays the foundation for the secondary and tertiary industries, the secondary industry drives the development of the primary industry, the primary and secondary industries create conditions for the tertiary industry, and the tertiary industry also promotes the development of the primary and secondary industries. Therefore, the output value of each industry and the proportion of GDP (gross domestic product) of each industry affect each other to varying degrees. The indicators of factors affecting power consumption are selected from the economic and social aspects. Consequently, the GDP per capita, the national total population, the urbanization level, the ratio of primary industrial output value to GDP, the ratio of secondary industrial output value to GDP, the ratio of tertiary industrial output value to GDP, and the total carbon emissions are selected as social and economic indicators. Finally, the gray correlation analysis model of three industry electricity consumption indicators and seven economic and social indicators is established.

Assuming that $H_i^{(0)}$ is a system factor, and its observed data on serial number $u$ is $h_i^{(0)}(u), i = 0, 1, \cdots, m, u = 1, 2, \cdots, n$, $H_i^{(0)} = \left\{ h_i^{(0)}(1), h_i^{(0)}(2), \cdots, h_i^{(0)}(n) \right\}$ is called the behavior sequence of factor $H_i^{(0)}$.

Assuming that $H_0{}^{(0)} = \left\{ h_0^{(0)}(1), h_0^{(0)}(2), \cdots, h_0^{(0)}(n) \right\}$ is the characteristic behavior sequence of the system,

$$
\begin{cases}
H_1^{(0)} = \left\{ h_1^{(0)}(1), h_1^{(0)}(2), \cdots, h_1^{(0)}(n) \right\} \\
\quad \vdots \\
H_i^{(0)} = \left\{ h_i^{(0)}(1), h_i^{(0)}(2), \cdots, h_i^{(0)}(n) \right\} \\
\quad \vdots \\
H_m^{(0)} = \left\{ h_m^{(0)}(1), h_m^{(0)}(2), \cdots, h_m^{(0)}(n) \right\}
\end{cases}
$$

is the sequence of relevant factors.

Then the grey relation coefficient is obtained by the following formula:

$$
\gamma(h_0^{(0)}(u), h_i^{(0)}(u)) = \frac{\underset{i}{\min}\,\underset{u}{\min}\left| h_0^{(0)}(u) - h_i^{(0)}(u) \right| + 0.5\underset{i}{\max}\,\underset{u}{\max}\left| h_0^{(0)}(u) - h_i^{(0)}(u) \right|}{\left| h_0^{(0)}(u) - h_i^{(0)}(u) \right| + 0.5\underset{i}{\max}\,\underset{u}{\max}\left| h_0^{(0)}(u) - h_i^{(0)}(u) \right|} \tag{1}
$$

The relation value can be calculated by

$$
\gamma(H_0^{(0)}, H_i^{(0)}) = \frac{1}{n}\sum_{u=1}^{n} \gamma(h_0^{(0)}, h_i^{(0)}), i = 1, 2, \cdots, m, \tag{2}
$$

$\gamma\left(H_0^{(0)}, H_i^{(0)}\right)$ is the correlation between the main behavior sequence and the related influenced factor sequence.

*3.2. Grey Correlation Analysis Model of Power Consumption in Three Major Industries*

Taking the electricity consumption of China's primary industry and the selected socioeconomic factors as an example, the following calculation is carried out according to the steps of gray correlation analysis:

The reference sequence is:

$$
H_0{}^{(0)} = (1014, 1003, 1027, 1014, 1040, 1093, 1175, 747, 779, \ 859)
$$

The comparison sequences are:

$$
H_1{}^{(0)} = (36277, 39771, 43497, 46912, 49922, 53783, 59592, 65534, 70078, 72000)
$$

$$
H_2{}^{(0)} = (134916, 135922, 136726, 137646, 138326, 139232, 140011, 140541, 141008, 141212)
$$

$$
H_3{}^{(0)} = (51.83, 53.10, 54.49, 55.75, 57.33, 58.84, 60.24, 61.50, 62.71, 63.89)
$$

$$
H_4{}^{(0)} = (9.2, 9.1, 8.9, 8.6, 8.4, 8.1, 7.5, 7.0, 7.1, 7.7)
$$

$$
H_5{}^{(0)} = (46.5, 45.4, 44.2, 43.1, 40.8, 39.6, 39.9, 39.7, 38.6, 37.8)
$$

$$
H_6{}^{(0)} = (44.3, 45.5, 46.9, 48.3, 50.8, 52.4, 52.7, 53.3, 54.3, 54.5)
$$

$$
H_7{}^{(0)} = (88.23, 90.00, 92.43, 92.88, 92.75, 92.74, 94.63, 96.49, 98.06, 98.94)
$$

From the steps in Section 3.1,

$$
\gamma(H^0{}_{(0)}, H^0{}_{(1)}) = 0.514 \qquad \gamma(H^0{}_{(0)}, H^0{}_{(2)}) = 0.715 \qquad \gamma(H^0{}_{(0)}, H^0{}_{(3)}) = 0.639
$$
$$
\gamma(H^0{}_{(0)}, H^0{}_{(4)}) = 0.771 \quad \gamma(H^0{}_{(0)}, H^0{}_{(5)}) = 0.758 \quad \gamma(H^0{}_{(0)}, H^0{}_{(6)}) = 0.648 \quad \gamma(H^0{}_{(0)}, H^0{}_{(7)}) = 0.679.
$$

By sorting the above data, it is found that: $0.771 > 0.758 > 0.715 > 0.679 > 0.648 > 0.639 > 0.514$. Therefore, the proportion of primary industry output value has the greatest correlation

with the primary industry power usage. According to this step, each reference sequence and corresponding comparison sequence of China's three major industries' power consumption are solved in turn, and the gray correlation degree obtained is shown in Table 3.

**Table 3.** Gray correlation analysis of China's socio-economic factors and power consumption of primary, secondary, and tertiary industries.

| Indicators Factors Electricity Consumption by Industry | Primary Industry Power Consumption | Electricity Consumption in the Secondary Industry | Electricity Consumption in the Tertiary Industry |
|---|---|---|---|
| Per capita GDP | 0.514 | 0.646 | 0.846 |
| Total population of the country | 0.715 | 0.677 | 0.603 |
| Urbanization level | 0.639 | 0.804 | 0.649 |
| Proportion of primary industrial structure | 0.771 | 0.522 | 0.527 |
| Proportion of secondary industrial structure | 0.758 | 0.562 | 0.560 |
| Proportion of tertiary industrial structure | 0.648 | 0.787 | 0.644 |
| Total carbon emissions | 0.679 | 0.723 | 0.621 |

Through Table 3, we can obtain the grey correlation analysis between power consumption of various industries and economic and social indicators. The results show that per capita GDP has the greatest correlation with electricity consumption of the tertiary industry, urbanization level has the greatest correlation with electricity consumption of the secondary industry, and the proportion of primary industry output value has the greatest correlation with its electricity consumption.

### 3.3. Grey Multivariable Convolution Model with Priority Accumulation of New Information (Abbreviated as GMCN(1,N))

According to the principle of new information first, the GMC(1,N) model can be increased by a parameter to change the weight of historical and new information, while cumulative generation is a more common method in grey system theory, which can more easily identify the regularity of the data [30]. The grey multivariable convolution model with priority accumulation of new information (GMCN(1,N)) is defined as follows: The GMCN(1,N) model is a grey prediction model with first-order differential equations and n variables.

(1) The cumulative generation operation using the new information priority principle (abbreviated as NAGO) is as follows:

From the original non-negative data, the original sequence is given:

$$X_i^{(0)} = \left\{ x_i^{(0)}(1), x_i^{(0)}(2), \cdots, x_i^{(0)}(m) \right\}, i = 1, 2, \ldots, N, \ m = 1, 2, \ldots, n. \tag{3}$$

$X_1^{(0)} = \left\{ x_1^{(0)}(1), x_1^{(0)}(2), \cdots, x_1^{(0)}(m) \right\}, m = 1, 2, \ldots, n$ is the original predicted data sequence, and $X_2^{(0)} = \left\{ x_2^{(0)}(1), x_2^{(0)}(2), \cdots, x_2^{(0)}(m) \right\}, m = 1, 2, \ldots, n$ is the sequence of influencing factors.

(2) The operation sequence (1-NAGO) generated by first-order new information priority accumulation on the original sequence is:

$$X_i^{(1)} = \left\{ x_i^{(1)}(1), x_i^{(1)}(2), \cdots, x_i^{(1)}(m) \right\}, \ i = 1, 2, \ldots, N. \tag{4}$$

where

$$x_i^{(1)}(k) = \sum_{j=1}^{k} \lambda^{k-j} x_i^{(0)}(j), i = 1, 2, \ldots, N, k = 1, 2, \ldots, m. \tag{5}$$

$\lambda$ can be adjusted according to the weight of old and new information, and we usually take $0 < \lambda < 1$. $X_i^{(1)}$ meets the new demand information priority principle, giving greater weight to newer data and smaller weight to older data.

(3) The whitening equation of this GMCN(1, N) model is written as

$$\frac{dx_1^1(t)}{dt} + b_1 x_1^{(1)}(t) = b_2 x_2^{(1)}(t) + b_3 x_3^{(1)}(t) + \ldots + b_N x_N^{(1)}(t) + u, (t = 1, 2, \ldots, m) \quad (6)$$

where the grey derivative of 1-NAGO is usually expressed as

$$\frac{dx_1^{(1)}(t)}{dt} = x_1^{(1)}(t+1) - x_1^{(1)}(t) \quad (7)$$

where $b_1, b_2, \ldots, b_N$ is a parameter, u is the ash consumption, and $\lambda$ can be adjusted according to the weight of the old and new information; we usually take $0 < \lambda < 1$.

(4) Calculated by the least square method,

$$\left[ \hat{b}_1, \hat{b}_2, \ldots, \hat{b}_N, \hat{u} \right]^T = (B^T B)^{-1} B^T Y \quad (8)$$

where

$$Y = \begin{bmatrix} x_1^{(1)}(2) - x_1^{(1)}(1) \\ x_1^{(1)}(3) - x_1^{(1)}(2) \\ \vdots \\ x_1^{(1)}(m) - x_1^{(1)}(m-1) \end{bmatrix} \quad (9)$$

$$B = \begin{bmatrix} -\frac{x_1^{(1)}(1) + x_1^{(1)}(2)}{2} & \frac{x_2^{(1)}(1) + x_2^{(1)}(2)}{2} & \cdots & \frac{x_n^{(1)}(1) + x_n^{(1)}(2)}{2} & 1 \\ -\frac{x_1^{(1)}(2) + x_1^{(1)}(3)}{2} & \frac{x_2^{(1)}(2) + x_2^{(1)}(3)}{2} & \cdots & \frac{x_n^{(1)}(2) + x_n^{(1)}(3)}{2} & 1 \\ \vdots & \vdots & \vdots & \vdots & \vdots \\ -\frac{x_1^{(1)}(m-1) + x_1^{(1)}(m)}{2} & \frac{x_2^{(1)}(m-1) + x_2^{(1)}(m)}{2} & \cdots & \frac{x_n^{(1)}(m-1) + x_n^{(1)}(m)}{2} & 1 \end{bmatrix} \quad (10)$$

The solved time response sequence is:

$$\hat{x}_1^{(1)}(t) = x_1^{(0)}(1) e^{-b_1(t-1)} + \sum_{\omega=2}^{t} \left\{ e^{-b_1(t-\omega+0.5)} \frac{f(\omega) + f(\omega-1)}{2} \right\} \quad (11)$$

where
$$f(t) = b_2 x_2^{(1)}(t) + b_3 x_3^{(1)}(t) + \cdots + b_N x_N^{(1)}(t) + u \quad (12)$$

(5) Through the cumulative generation operator, the reduction sequence is:

$$\hat{x}_1^{(0)}(1) = x_1^{(0)}(1), \hat{x}_1^{(0)}(k) = \hat{x}_1^{(1)}(k) - \lambda \hat{x}_1^{(1)}(k-1), k = 2, 3, \ldots, m \quad (13)$$

$\hat{x}_1^{(0)}(k)$ is the predicted value of the original sequence

When $\lambda = 1$, the GMCN(1,N) model is the traditional GMC(1,N) model. The GMCN(1,N) model makes full use of new information.

(6) In this paper, the mean absolute percentage error (MAPE) is used to evaluate the fitting accuracy of the model, where

$$\text{MAPE} = 100\% * \frac{1}{m} \sum_{t=1}^{m} \left| \frac{\hat{x}_1^{(0)}(t) - x_1^{(0)}(t)}{x_1^{(0)}(t)} \right| \quad (14)$$

After consulting the data, the evaluation criteria of MAPE are determined, as shown in Table 4 [31].

**Table 4.** The model evaluated classification of MAPE value.

| MAPE | Prediction Effect |
| --- | --- |
| 0~10% | Excellent |
| 10~20% | Good |
| 20~50% | General |
| >50% | Poor |

To facilitate the understanding of the solution process in this paper, all models and logical relations involved in this paper are presented in a frame diagram, as shown in Figure 1.

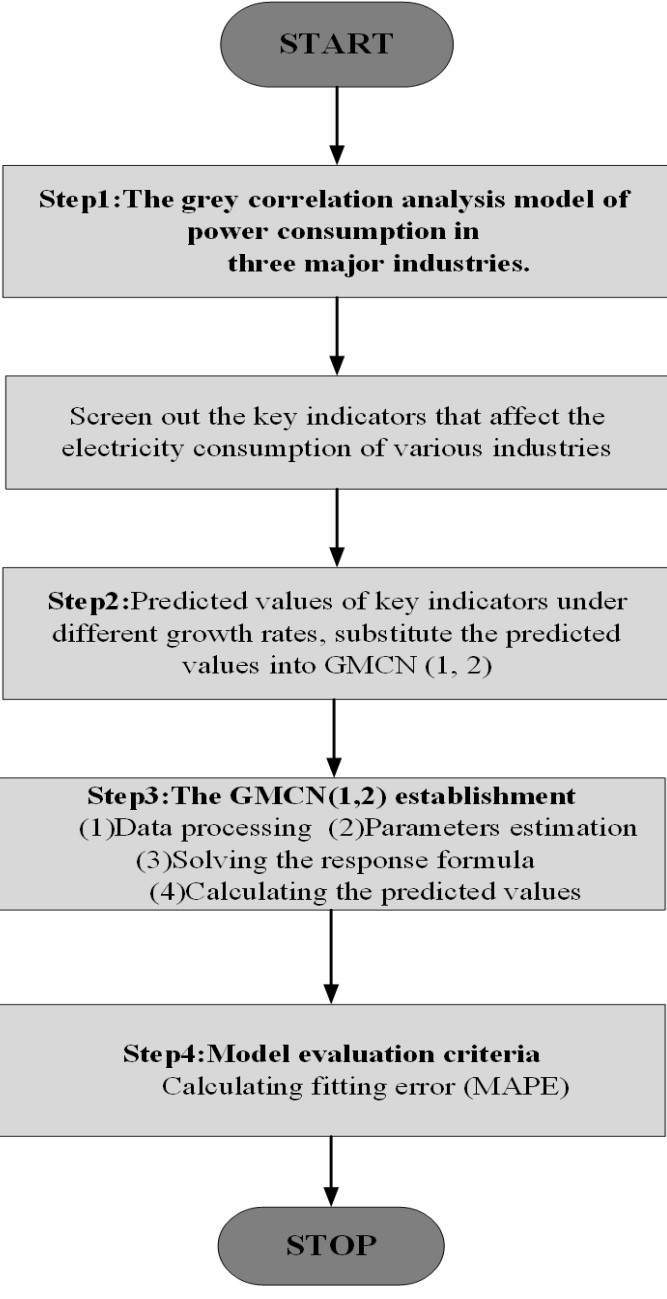

**Figure 1.** The process of three major industrial power forecasting.

## 4. Using GMCN(1,2) to Predict China's Tertiary Industry Electricity Consumption under Different per Capita GDP Growth Rates

Although China is currently known as the "world's factory", the secondary industry has retreated to second place in the economic share, and the tertiary industry has become the industry with the highest output value in China. Accordingly, the tertiary industry electricity consumption in China is also rising year by year. Thus, we analyze the tertiary industry power consumption, the secondary industry power consumption, and the primary industry power consumption in turn. Other than the main and secondary sectors, the tertiary industry includes information transmission, postal services, information transmission, wholesale and retail, lodging and catering, finance and real estate, computer and software services, etc.

Through the grey correlation analysis model in Table 3, it can be concluded that the association between the per capita GDP and power consumption of the tertiary industry is the strongest. Since the future trend of per capita GDP is uncertain, it is assumed that the per capita GDP growth rate is −5%, 5%, 10%, and 15%, respectively. The future consumption of electricity in the tertiary industry are observed under these four different scenarios. There are three reasons to set such a percentage increase. The first reason is that from 2011 to 2020, the yearly average growth rate of China per capita GDP was 8%. At the same time, there are many emerging industries in China. These industries are developing rapidly, and their development potential is unpredictable. China is in the developing stage, and the future development level is uncertain. Therefore, four scenarios of per capita GDP growth rates of −5%, 5%, 10%, and 15% are considered. The second reason is that the impact of the new crown pneumonia epidemic and changes in the global development pattern affect the changes in per capita GDP to a certain extent. The third factor is if there is an inflection point in the relationship between GDP per capita and power consumption in the tertiary industry. In other words, when the GDP per capita growth rate reaches this inflection point, it is an interesting question whether it will continue to promote the rise of electricity consumption in the tertiary industry. To explore this question, the per capita GDP growth rates are assumed to −5%, 5%, 10%, and 15%, respectively, and the corresponding per capita GDP from 2021 to 2030 can be calculated and obtained. Finally, the calculation results in Table 5 are brought into the GMCN(1,2) model, and the future power consumption of the tertiary industry may be forecasted under four different scenarios. The fitting results of the GMCN(1,2) model are shown in Figure 2. The MAPE value of the GMCN(1,2) model is 2.79%, which is less than 10%, indicating that the model fits well.

**Table 5.** Per capita GDP at different growth rates.

| Per Capita GDP (Yuan) | 2021 | 2022 | 2023 | 2024 | 2025 | 2026 | 2027 | 2028 | 2029 | 2030 |
|---|---|---|---|---|---|---|---|---|---|---|
| −5% | 68,400 | 64,980 | 61,731 | 58,644 | 55,712 | 52,927 | 50,280 | 47,766 | 45,378 | 43,109 |
| 5% | 75,600 | 79,380 | 83,349 | 87,516 | 91,892 | 96,487 | 101,311 | 106,377 | 111,696 | 117,280 |
| 10% | 79,200 | 87,120 | 95,832 | 105,415 | 115,957 | 127,552 | 140,308 | 154,338 | 169,772 | 186,749 |
| 15% | 82,800 | 95,220 | 109,503 | 125,928 | 144,818 | 166,540 | 191,521 | 220,250 | 253,287 | 291,280 |

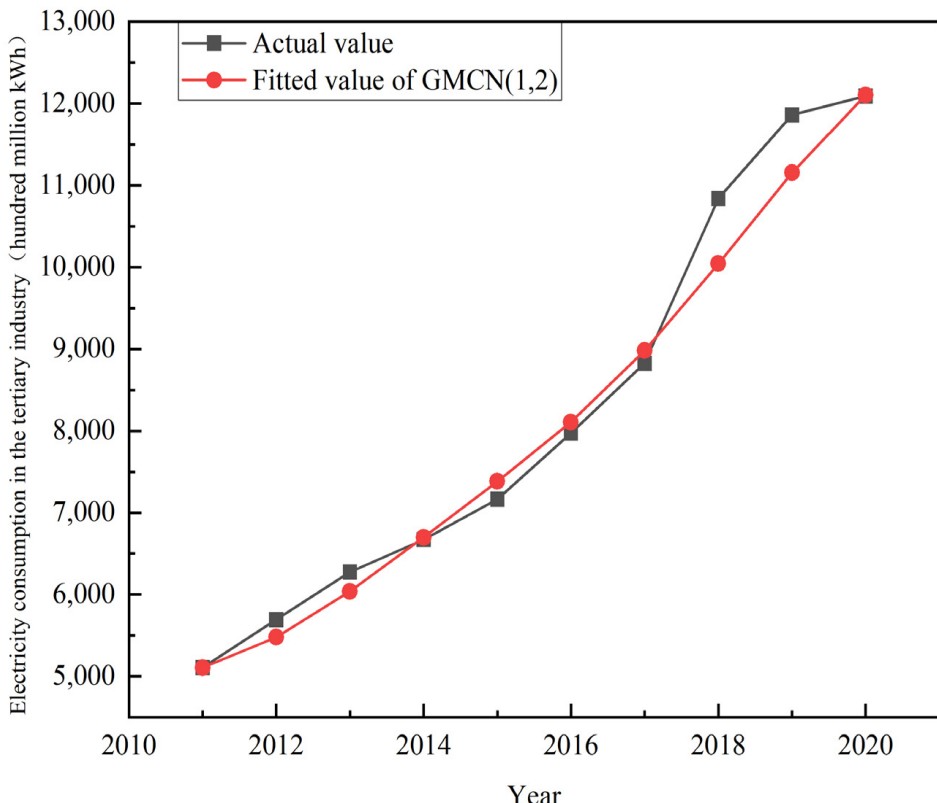

**Figure 2.** Fitting between predicted and actual values.

The anticipated power consumption values for the tertiary industry under different growth rates are shown in Table 6, and Figure 3 depicts the trend chart of power usage in the tertiary industry under four different scenarios. In Figure 3, with an increase in GDP per capita, the tertiary industry's electricity demand rises, and the energy supply and demand become larger. If the per capita GDP is reduced by 5%, the tertiary industry's power demand will also decrease. China's tertiary industrial power demand is anticipated to reach 815.8 billion kWh in 2030, which will be less than one trillion kWh. If the situation of scenario 1 occurs, with the development of the country, it is possible to meet such electricity demand. However, China is currently in a developing period, and there is still a long way to go from developed countries. With the reorganization of the industrial structure and the expansion of the economy, the per capita GDP will naturally increase. When per capita GDP increases by 5%, 10%, and 15%, respectively, the anticipated values of the tertiary industry's electricity demand in 2030 will be 2158.5 billion kWh, 3303.6 billion kWh, and 4929.2 billion kWh, respectively. From Figure 3, we can find that when the GDP per capita progressively rises, the demand for electricity in the tertiary industry also rises. According to the growth trend of the country's per capita GDP from 2011 to 2020, it is highly likely that China's tertiary sector power demand will exceed three trillion kWh in 2030. Of course, this is an estimate under conservative circumstances. Because of the continuous development of society, China's per capita GDP has continued to grow from past development. When the GDP per capita growth rate is 15%, the electricity demand of the country's tertiary industry will reach 4.9 trillion kWh. According to the GDP per capita growth pattern in recent years, the possibility of scenario 4 occurring is relatively small. However, summarizing the four hypothetical scenarios, the tertiary industry's power usage will continue to rise, which will pose a greater challenge to China's power supply and demand capabilities.

**Table 6.** Predicted power consumption of the tertiary industry under different growth rates.

| Per Capita GDP Growth Rate | 2021 | 2022 | 2023 | 2024 | 2025 | 2026 | 2027 | 2028 | 2029 | 2030 |
|---|---|---|---|---|---|---|---|---|---|---|
| −5% | 12,574 | 12,516 | 12,168 | 11,666 | 11,092 | 10,489 | 9883 | 9288 | 8712 | 8158 |
| 5% | 12,914 | 13,733 | 14,573 | 15,444 | 15,352 | 17,301 | 18,297 | 19,340 | 20,435 | 21,585 |
| 10% | 13,084 | 14,366 | 15,894 | 17,644 | 19,608 | 21,791 | 24,206 | 26,870 | 29,805 | 33,036 |
| 15% | 13,254 | 15,017 | 17,299 | 2,0071 | 23,345 | 27,159 | 31,575 | 36,669 | 42,538 | 49,292 |

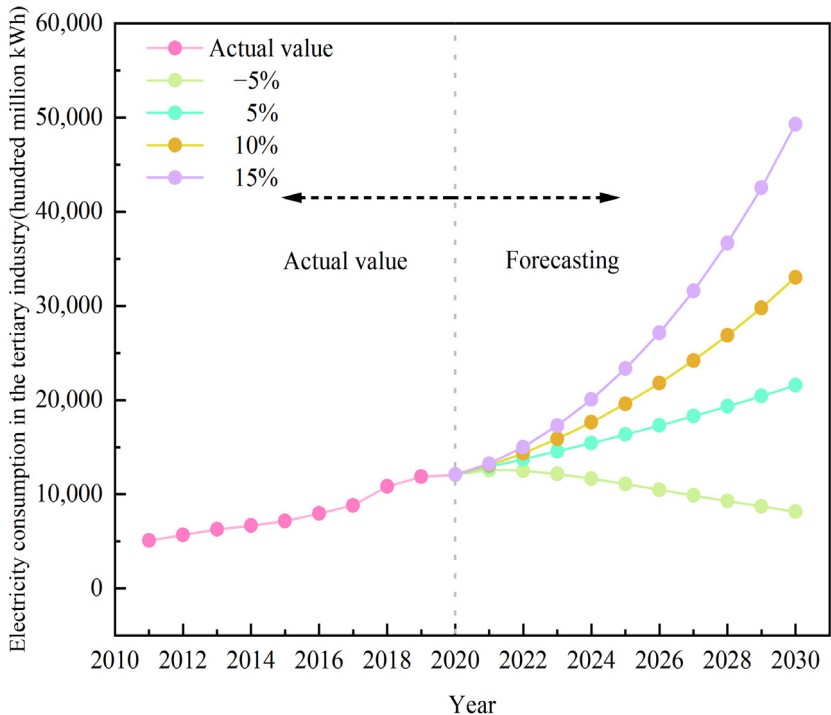

**Figure 3.** Forecast results of power consumption of China's tertiary industry.

## 5. Using GMCN(1,2) to Predict the Power Consumption of China's Secondary Industry under Different Urbanization Growth Rates

The secondary industry includes mining, manufacturing, power, construction, gas, and water production and supply. The rapid growth of exports, the sustained development of industrial production, and the rapid growth of high-tech manufacturing and equipment manufacturing have caused the increase in the secondary industry's electricity demand. With the significant acceleration of the growth rate of industrial power consumption such as high-tech and equipment manufacturing industries, the power consumption of high energy consuming industries has increased rapidly, and the growth rate of power consumption is faster than the planning expectation, which puts forward a greater challenge on China's power generation capacity.

Through the gray correlation analysis model in Table 3, it can be concluded that the urbanization level has the strongest correlation with secondary industry electricity consumption. Since the future trend of the urbanization level is uncertain, it is assumed that the growth rate of the urbanization level is −1%, 1%, 2%, and 3%, respectively. There are two reasons to set such percentage increases. The first reason is that the average annual growth rate of the urbanization level in China from 2011 to 2020 was 2%. The second reason is that with the vigorous promotion of the rural revitalization strategy, the population transferred to cities will be less and less, the level of urbanization will tend to be stable, and large increases are unlikely. Therefore, four scenarios of urbanization level growth rate of −1%, 1%, 2%, and 3% are considered. When the urbanization level growth rates are

assumed to be −1%, 1%, 2%, and 3%, respectively, the urbanization levels from 2021 to 2030 can be calculated. Table 7 demonstrates the results. Finally, the calculation results in Table 7 are brought into the GMCN(1,2) model to obtain the anticipated value of the secondary industry's future power usage under four different scenarios. The predicted results are displayed in Table 8. Figure 4 displays the results of the GMCN(1,2) model's fitting. The MAPE value of the GMCN(1,2) model is 1.36%, which is less than 10%, indicating that the model fitting effect is excellent.

**Table 7.** Urbanization level under different growth rates.

| Urbanization Level Growth Rate | 2021 | 2022 | 2023 | 2024 | 2025 | 2026 | 2027 | 2028 | 2029 | 2030 |
|---|---|---|---|---|---|---|---|---|---|---|
| −1% | 60.70 | 57.66 | 54.78 | 52.04 | 49.44 | 46.97 | 44.62 | 42.39 | 40.27 | 38.25 |
| 1% | 64.53 | 65.17 | 65.83 | 66.48 | 67.15 | 67.82 | 68.5 | 69.18 | 69.88 | 70.57 |
| 2% | 65.17 | 66.47 | 67.80 | 69.16 | 70.54 | 71.95 | 73.39 | 74.86 | 76.35 | 77.88 |
| 3% | 65.81 | 67.78 | 69.81 | 71.91 | 74.07 | 76.29 | 78.58 | 80.93 | 83.36 | 85.86 |

**Table 8.** Predicted power consumption of the secondary industry under different growth rates.

| Urbanization Level Growth Rate | 2021 | 2022 | 2023 | 2024 | 2025 | 2026 | 2027 | 2028 | 2029 | 2030 |
|---|---|---|---|---|---|---|---|---|---|---|
| −1% | 52,983 | 54,344 | 55,422 | 56,241 | 56,825 | 57,195 | 57,371 | 57,371 | 57,214 | 56,913 |
| 1% | 53,122 | 54,889 | 56,615 | 58,302 | 59,952 | 61,568 | 63,152 | 64,707 | 66,233 | 67,733 |
| 2% | 53,145 | 54,982 | 56,822 | 58,668 | 60,520 | 62,382 | 64,253 | 66,137 | 68,034 | 69,945 |
| 3% | 53,169 | 55,075 | 57,031 | 59,040 | 61,102 | 63,221 | 65,397 | 67,633 | 69,931 | 72,292 |

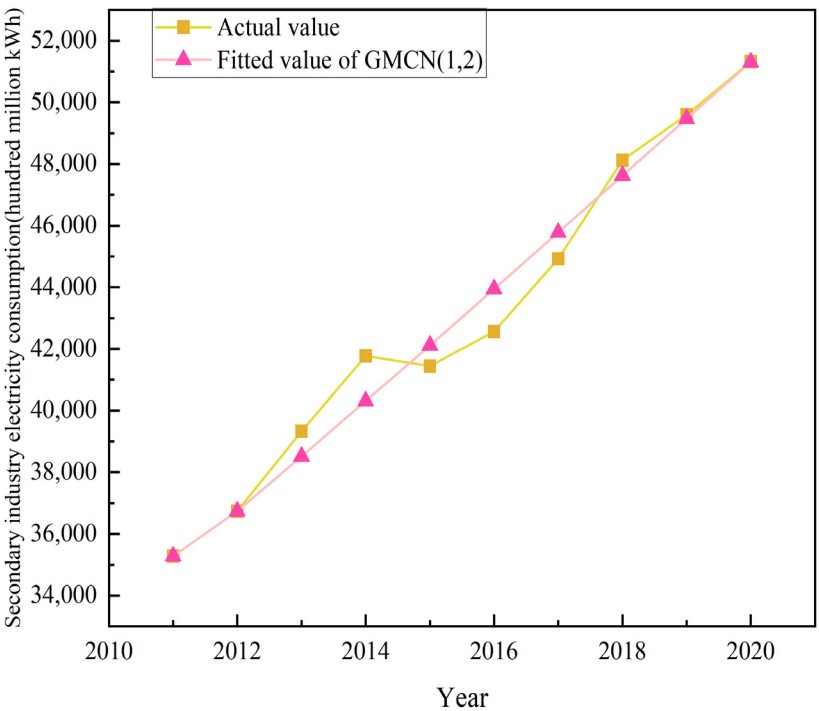

**Figure 4.** Fitting between predicted and actual values.

The trend chart of secondary industrial power usage under the four scenarios is shown in Figure 5. In Figure 5, as the level of urbanization increases, the secondary industry's power usage likewise rises. If the level of urbanization is reduced by 1% on average, the secondary industry's power usage will first rise to 5737.1 billion kWh, reaching an inflection point, and then will decrease subsequently. This indicates that with the development of the

social economy, the balance between urban and rural development has been reached, the structure of industry has been modified, and the secondary industrial electricity demand has declined accordingly. When the urbanization level increases by 1%, 2%, and 3%, respectively, by 2030, the predicted value of electricity demand in China's secondary industry will be 6773.3, 6994.5, and 7229.2 billion kWh, respectively. Through Figure 5, we can observe the trend of future electricity demand in China's secondary industry under different scenarios. According to the growth trend of urbanization level from 2011 to 2020, China's power demand of secondary industry will reach 7 trillion kWh in 2030. Although China's industrial structure is undergoing adjustment and development, the secondary industry's power usage still needs our attention. On the basis of completing the carbon peaking goal, meeting the power needs of various industries has greater requirements for China's policy guidance and technological innovation.

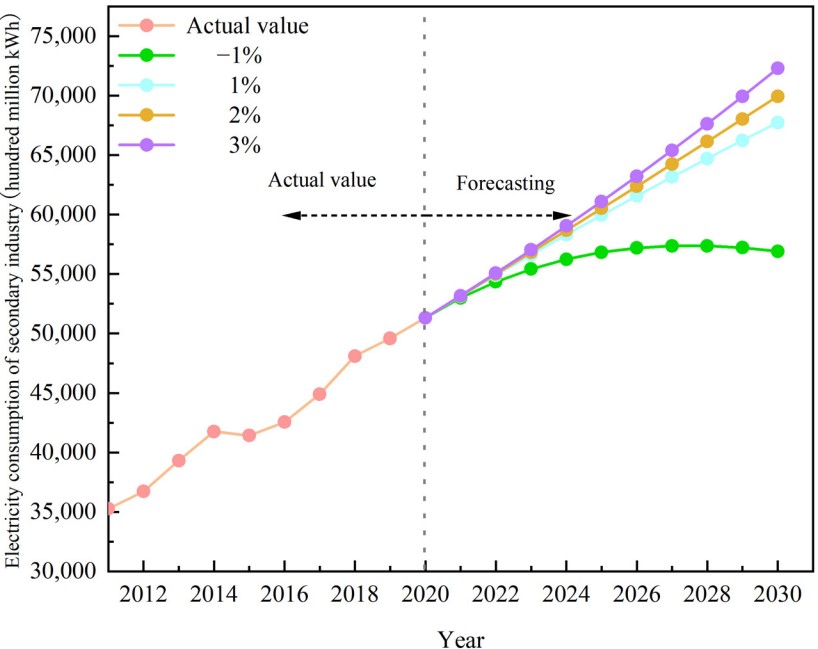

**Figure 5.** Forecasted results of power consumption of China's secondary industry.

## 6. Using GMCN(1,2) to Predict the Primary Industry's Power Consumption under Different Growth of the Primary Industry Output Value Ratio

Primary industry activities, which include agriculture, forestry, animal husbandry, and fishing, all contribute to the overall power consumption of the primary industry. The primary industry output value in GDP and China's primary industrial power usage fluctuated greatly, showing a nonlinear relationship. Because the GMCN(1,2) model is capable of studying nonlinear relationships and small datasets, it is reasonable to use the GMCN(1,2) model to analyze this set of data. Through the gray correlation analysis model in Table 3, it can be concluded that the association between the proportion of the primary industry output value and the primary industrial power usage is the strongest. Nowadays, China's three major industrial structures are in the pattern of "three, two, and one", and are moving towards a stage of high-efficiency comprehensive development. The primary industry accounts for less than 10%. Although agriculture is the foundation of the country, its output value is already relatively low. Therefore, the future trend of the primary industrial GDP proportion is uncertain. It is assumed that the growth rate of the primary industry output value in GDP is −4%, −2%, 2%, and 4%, respectively. The future consumption of electricity in the primary industry can be observed under these four different scenarios. There are two reasons to set such a percentage increase. The first reason is that the average growth rate of the primary industry output value in GDP in China from

2013 to 2020 is −2%. The expansion of the economy will boost the GDP of the three major industries, but the primary industrial GDP growth rate may not be as high as that of the secondary and tertiary industries. The primary industry output value in GDP is likely to decrease. The second reason is that the new COVID-19 pandemic has had a negative impact on the secondary and tertiary industries, while the impact on the primary industry has been positive. The pandemic has existed for a long time, so the primary industry output value in GDP is anticipated to increase. When it is assumed that the growth rate of the primary industry output value in GDP is −4%, −2%, 2%, and 4%, respectively, the share of GDP attributable to the primary industry from 2021 to 2030 can be calculated and acquired. Table 9 demonstrates the results. Finally, the calculation results in Table 9 are brought into the GMCN(1,2) model to obtain the predicted value of the primary industrial power consumption under four different scenarios. The predicted results are displayed in Table 10. The fitting results of the GMCN(1,2) model are displayed in Figure 6. The GMCN(1,2) model's MAPE value is 5.78%, which is less than 10%, indicating that the model fitting effect is excellent.

**Table 9.** Proportion of primary industry GDP under different growth rates.

| Proportion of Primary Industry GDP | 2021 | 2022 | 2023 | 2024 | 2025 | 2026 | 2027 | 2028 | 2029 | 2030 |
|---|---|---|---|---|---|---|---|---|---|---|
| −4% | 7.39 | 7.10 | 6.81 | 6.54 | 6.28 | 6.03 | 5.79 | 5.55 | 5.33 | 5.12 |
| −2% | 7.55 | 7.40 | 7.25 | 7.10 | 6.96 | 6.82 | 6.68 | 6.55 | 6.42 | 6.29 |
| 2% | 7.85 | 8.01 | 8.17 | 8.33 | 8.50 | 8.67 | 8.84 | 9.02 | 9.20 | 9.39 |
| 4% | 8.01 | 8.33 | 8.66 | 9.01 | 9.37 | 9.74 | 10.13 | 10.54 | 10.96 | 11.40 |

**Table 10.** Predicted power consumption of the primary industry under different growth rates.

| Growth Rate of Primary Industry in GDP | 2021 | 2022 | 2023 | 2024 | 2025 | 2026 | 2027 | 2028 | 2029 | 2030 |
|---|---|---|---|---|---|---|---|---|---|---|
| −4% | 810 | 774 | 706 | 623 | 533 | 443 | 353 | 265 | 179 | 97 |
| −2% | 824 | 819 | 792 | 752 | 707 | 659 | 610 | 561 | 513 | 465 |
| 2% | 849 | 909 | 968 | 1026 | 1085 | 1146 | 1206 | 1268 | 1332 | 1397 |
| 4% | 862 | 956 | 1061 | 1174 | 1294 | 1420 | 1551 | 1688 | 1831 | 1980 |

Although the share of power consumed by the primary sector is less than that of the secondary and tertiary industries, the power usage in the primary industry has increased rapidly, driven by factors such as a new record of grain output, the stable growth of animal husbandry production, and the continuous improvement of agricultural electrification. The continuous improvement of rural power consumption conditions, coupled with the continuous and in-depth promotion of poverty alleviation, have driven rural development and jointly promoted the revelation of primary industry's power consumption potential. According to the predicted value, we should sensibly alter the industrial layout and improve the power supply level.

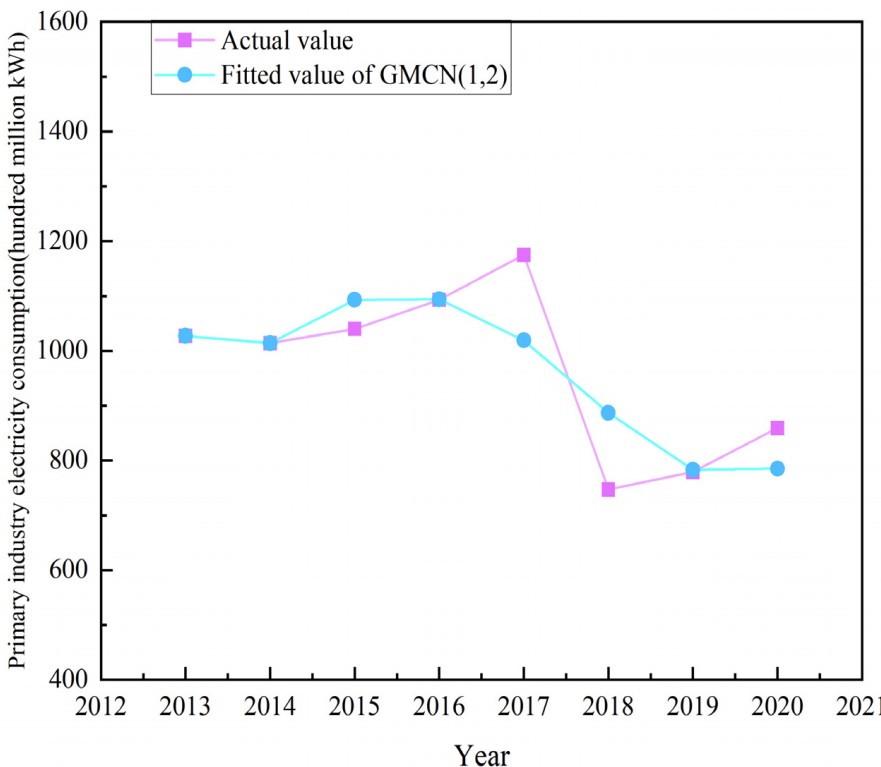

**Figure 6.** Fitting between predicted and actual values.

As seen in Figure 7, with the decline of the proportion of primary industry output value, the primary industry's energy consumption demand is likewise steadily reduced. However, with the improvement of the technical level of the industry, the primary industrial power demand still shows a slight upward trend. Even though the power requirements of the primary industry are a fraction of those of the secondary and tertiary industries, against the background of the global COVID-19 pandemic, primary industry power consumption is expected to continue its recent rising trend. The power requirement of the primary industry is analyzed and forecasted under four different scenarios. When growth rates of the primary industry output value in GDP are −4% and −2%, the power demand of the primary industry will be 9.7 billion kWh and 46.5 billion kWh, respectively, by 2030. Obviously, when growth rate of the primary industry output value in GDP in the annual average drops by 4%, the power demand of the primary industry will be too small to be realistic. Therefore, it is expected that the share of GDP generated by the primary industry output value will increase by 1%, and the primary industry power usage will develop steadily. When the growth rates of primary industry output values in GDP are 2% and 4%, the electricity requirement of the primary industry will be 139.7 billion kWh and 198 billion kWh, respectively, by 2030. Although the industrial structure has been regularly modified, the primary industry output value share of GDP will have increased over the previous two years. The power requirement of the primary industry is expected to reach 150 billion kWh in 2030.

As a part of terminal energy, the change of demand for electric energy is affected by the energy demand layout. Therefore, it is very important to reasonably and accurately predict the consumption of electricity. According to the predicted value, we should reasonably adjust the industrial layout, improve the power supply level, strive to meet the industrial power demand, and improve the national economy under the goal of achieving the carbon peak.

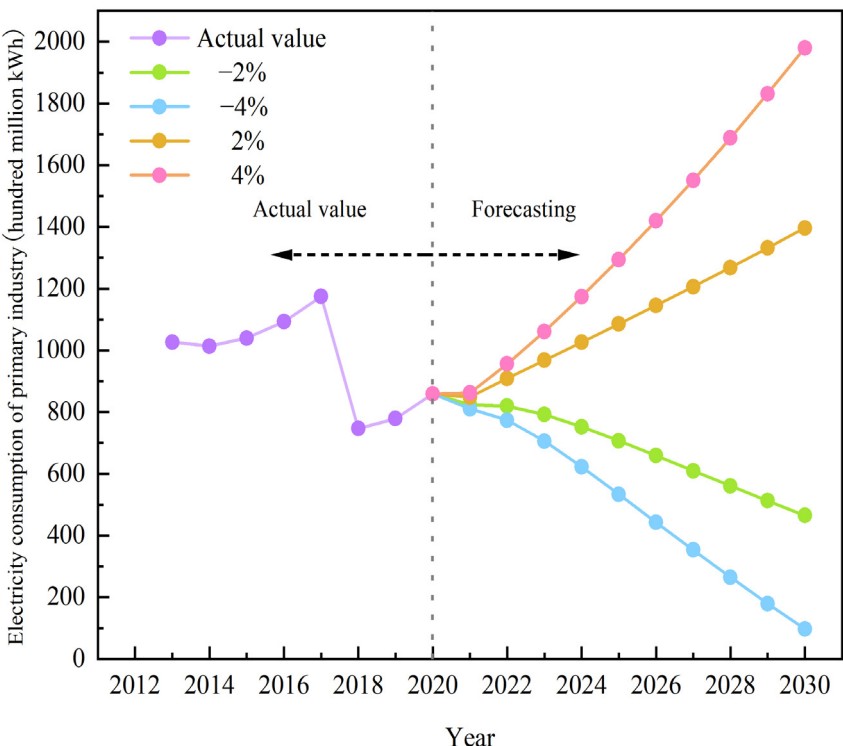

**Figure 7.** Power consumption forecast results of China's primary industry.

## 7. Conclusions and Prospect

This paper uses the GMCN(1,N) model to predict the electricity consumption of China's three major industries. The results show that China's electricity consumption is still on the rise. Due to the uncertainty of future development, we separately discuss the electricity consumption of the three major industries under different scenarios. The factor most closely related to the power consumption of the tertiary industry is GDP per capita. When the growth rate of GDP per capita is −5%, 5%, 10%, and 15%, the electricity demand value of China's tertiary industry will reach 815.8 billion kWh, 2158.5 billion kWh, 3303.6 billion kWh, and 4929.2 billion kWh, respectively, in 2030. The relationship between the urbanization level and the electricity consumption of the secondary industry is the closest. When the growth rate of the urbanization level is −1%, 1%, 2%, and 3%, the electricity demand value of China's secondary industry will reach 5737.1 billion kWh, 6773.3 billion kWh, 6994.5 billion kWh, and 7229.2 billion kWh, respectively, in 2030. The GDP ratio of the primary industry affects the electricity consumption of the primary industry to a large extent. When the growth rate of primary industry output value in GDP is −4%, −2%, 2%, and 4%, the electricity consumption of the primary industry will reach 9.7 billion kWh, 46.5 billion kWh, 139.7 billion kWh, and 198 billion kWh, respectively, in 2030. If China continues to develop at the growth rate from 2011 to 2020, it is estimated that by 2030, the electricity usage of the tertiary industry will reach three trillion kilowatt hours, the secondary industry's power usage will reach seven trillion kilowatt hours, and the primary industry's electricity consumption will reach 150 billion kWh. Therefore, it is expected that the total power consumption of China's three major industries will reach 10.15 trillion kWh. When the development trend develops and changes, we can predict the scale and development trend of power consumption in various industries in advance by observing the changes of key indicators and combining the different scenarios presented in this paper. Judging from the power consumption results of China's three major industries, the power consumption of the secondary industry and the tertiary industry accounts for a large proportion, the growth rate of the power consumption of the secondary industry has slowed down, and the growth trend of the power consumption of the tertiary industry is

obvious. We suggest that the government adjust the industrial structure and build a smart grid according to the power consumption trends of different industries so as to formulate active and effective energy policies.

According to the characteristics of China's power generation energy resources and the distribution of power load, we should coordinate economic and social development. Accelerating a change in power development mode should be a priority for power development, always giving priority to conservation and to the development of hydropower, actively and in an orderly manner developing new energy power generation, and promoting the optimal allocation of power resources on a larger scale. It is necessary to accelerate the construction of a smart grid and build a safe, economic, green, and harmonious modern power industry system so as to fulfil the future power requirements of the economy and society to the greatest extent and to promote the efficient development of society.

**Author Contributions:** Conceptualization, methodology, investigation, writing—original draft, review and editing, project administration, Y.X.; conceptualization, review and editing, visualization, project administration, L.W.; software, formal analysis, investigation, data curation, writing—review and editing, Y.Y. All authors have read and agreed to the published version of the manuscript.

**Funding:** The relevant researches are supported by the National Natural Science Foundation of China (71871084, U20A20316), the Young talent support scheme of Hebei Province (360-0803-YBN-7U2C), the key research project in humanity and social science of Hebei Education Department (ZD202211), the Natural Science Foundation of Hebei Province (E2020402074), and the Social Science Federation Project of Handan (2022072, 2022088).

**Data Availability Statement:** China's three major industries' economic and social data are all from China's statistical yearbook. The electricity consumption data of China's three major industries come from China Electricity Statistical Yearbook. We put China's economic and social data from 2011 to 2020 on github, and the usage data can be downloaded through the link https://github.com/han616807/hello (accessed on 26 July 2022).

**Conflicts of Interest:** The authors declare no conflict of interest.

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
