# Peer review of "Power Consumption Forecast of Three Major Industries in China Based on Fractional Grey Model"

_axioms, doi:10.3390/axioms11080407_

Round 1

Reviewer 1 Report

The article concerns the forecast of electricity demand in the perspective of years. This forecast is made using the fractional grey model. This model is used to find a forecast for China separately for the primary , secondary and tertiary  industry. The model used is taken from the literature. The article is not of a scientific nature. It is technical in nature.

Author Response

Detailed Response to Reviewer 

Firstly, thanks for your careful and insightful comments. The revisions are listed as follows:

Reviewer 1:

  1. Does the introduction provide sufficient?

We reorganized the background, questions, methods and significance of the research. The introduction is more clearly expressed in Section 1 of this paper.

  1. Are all the cited references relevant to the research?

The References section has been restructured. Some references have been removed and some new ones have been added. Specifically as follows.

As the foundation and guarantee of social development, energy has very important economic and social value[1]. With the rapid economic and social development, the energy problem has become increasingly prominent, and the energy consumption and future development trends have attracted extensive attention from scholars at home and abroad. Xue Yuexin et al. predicted Beijing's total energy consumption will reach 85.728 million tons in 2025, providing reference and guidance for Beijing's energy development planning[2]. Sun Manli et al. predicted that in 2025, the total installed capacity of natural gas distributed energy in China will reach 33.655 million kilowatts, 36.385 million kilowatts and 40.25 million kilowatts respectively under the high, medium and low scenarios[3]. Jiao Jianling predicted China's future oil demand in the context of carbon emission reduction[4]. Wang Bing et al. estimated that China's coal consumption will reach 560 million to 760 million tce in 2060, which will help guide the development trend of coal energy[5].

With the continuous acceleration of urbanization and industrialization in China, the rapid economic and social development has increased the demand for energy, especially electric energy. Electricity will be the mainstay of energy growth in the next 10 years[6]. Maswanganyi et al. used a partial linear additive quantile regression model to forecast electricity consumption in South Africa[7]. Al-Musaylh et al. proved that the MARS and SVR models are more suitable for the Queensland electricity demand study than the ARIMA model[8]. Yu Xiaoxiao used gray forecasting and time trend extrapolation to longitudinally analyze the electricity consumption data in Anhui Province, made short-term forecasts for electricity demand [9]. Wu et al. established a new multivariate grey prediction model considering the total population to predict the electricity consumption in Shandong Province[10]. Ren Fangling et al. selected the annual electricity consumption in Shaanxi Province as the research object, and achieved good prediction results through the optimized multiple linear regression model and gray GM model[11]. Nie Jing uses the LMDI model to explore the power consumption intensity of Beijing-Tianjin-Hebei[12]. Su Zhenyu established a multiple linear regression equation to forecast the electricity demand in Suzhou[13]. Li Wencong used the optimized GM(1,1) model to explore and analyze the electricity consumption of the whole society in Hubei Province[14]. Chen Tingting analyzed the terminal energy consumption in Shaanxi Province by improving the variable weight combination forecasting model, and provided suggestions for the development of the electricity market in Shaanxi Province[15]. Zhai Ying et al. analyzed the relationship between the cyclical fluctuation of per capita electricity consumption and various influencing factors in various regions of China[16]. Williams et al. used a piecewise interpolation operation on the time series of discrete observations, developed a power demand forecasting method, and provided a reliable forecasting method for short-term power consumption[17]. Lebotsa proposed a mixed model of bounded variables to predict short-term electricity consumption[18]. Based on the time series decomposition method and regression analysis method, Wang Yanbo proposed a comprehensive forecasting method for monthly electricity consumption based on the STL decomposition model[19]. Hongye Guo et al. proposed a monthly electricity consumption prediction framework based on a vector correction model, and explored the potential connection of energy footprint[20]. Xiao Zheng et al. based on grey forecasting and time series methods, respectively constructed grey Verhulst-ARIMA forecasting and seasonal ARIMA models to forecast seasonal electricity consumption[21]. Gao Hong et al. used the GM(1,1) model to predict and analyze the power consumption of China's manufacturing industry, and put forward corresponding suggestions[22]. Ma Chao et al. took Tianjin's manufacturing industry as the main body, and made predictions after analyzing the development trend of electricity consumption, which provided a reference for the construction of Tianjin power grid[23]. Liu Ruiyu et al. used the scenario analysis method to study the medium and long-term development potential of electricity consumption in China's high-energy-consuming industries under seven scenarios, indicating that high-energy-consuming industries still have a great pulling effect on the economy[24]. Qu Bo et al. analyzed the demand of energy and electricity consumption side under the background of "carbon neutrality", explored the development direction of energy green consumption transformation, and explored new development models of electricity consumption side from different fields such as industry, construction, transportation, and agriculture[25].

  1. Is the research design appropriate?

In this paper, seven factors closely related to power consumption are selected for grey correlation analysis, and the key indicators that affect the power consumption of various industries are screened out. Then, the predicted values of the impact indicators under different scenarios are obtained. Finally, the GMCN (1, 2) models of the three major industries are established to predict the development trend of power consumption in the three major industries.

The reasons for applying the grey relational analysis model and the grey multivariate model with the priority of new information to study the electricity consumption of the three major industries are as follows: compared with the traditional mathematical statistics method, the gray relational analysis model has fewer observations, and does not require too much sample size, nor does it require typical distribution laws, the calculation is simple, and the quantitative analysis results will be more consistent with the qualitative analysis results. The relevant factors affecting the main factors are objectively analyzed by using the grey relational analysis method. It can explore uncertain relationships between one major factor of the system and other factors. Electricity consumption is affected by various external factors to varying degrees, which are related to factors such as population, economy, industry, and ecology. These complex factors are not completely clear in a certain period of time, we can understand it as a gray system, and use the gray correlation analysis model to explore the relationship between the electricity consumption of the three major industries and other factors in the gray system. Therefore, the grey correlation model is used to calculate the correlation between different influencing factors in China's economy and society and the electricity consumption of the three major industries. In this paper, the grey multivariate model with the priority of new information is used to forecast the electricity consumption of the three major industries respectively. New information-first grey multivariate models have two advantages over traditional forecasting models. First, the model considers the priority of new information, and gives greater weight to the new information in the calculation process, which can make future predictions more accurate. Secondly, the model takes into account multiple variables, not just the forecast of a single electricity consumption. On the basis of the normal development trend, the grey multivariate model takes into account the index with the greatest correlation with electricity consumption, making the forecast more realistic . Through observation, we found that the MAPE values of the GMCN (1, 2) models of power consumption in the primary, secondary, and tertiary industries were 5.78%, 1.36%, and 2.79%, all less than 10%. The fitting effect is very good, so it can be used to predict the power consumption of the three major industries.

Therefore, the research design of this paper is suitable for solving the problem of forecasting power consumption in China's three major industries.

  1. Are the methods adequately described?

The research method is more clearly expressed in Section 3 of this paper. In order to facilitate understanding, the following simple flow chart is drawn in figure 1.

 Figure 1. The process of three major industrial power forecasting.

  1. Are the results clearly presented?

The last part of the article reorganizes the results, as follows:

This paper uses the GMCN (1, N) model to predict the electricity consumption of China's three major industries. The results show that China's electricity consumption is still on the rise. Due to the uncertainty of future development, we separately discussed the electricity consumption of the three major industries under different scenarios. The factor most closely related to the power consumption of the tertiary industry is GDP per capita. When the growth rate of GDP per capita is -5%, 5%, 10% and 15% respectively, the electricity demand value of China's tertiary industry will reach 815.8 billion kWh, 2158.5 billion kWh, 3303.6 billion kWh, 4929.2 billion kWh in 2030. The relationship between the urbanization level and the electricity consumption of the secondary industry is the closest. When the growth rate of the urbanization level is -1%, 1%, 2%, and 3%, the electricity demand value of China's secondary industry will reach 5,737.1 billion kWh, 6,773.3 billion kWh, 6,994.5 billion kWh, and 7,229.2 billion kWh in 2030. The GDP ratio of the primary industry affects the electricity consumption of the primary industry to a large extent. When growth rate of primary industry output value in GDP is -4%, -2%, 2%, and 4%, the electricity consumption of the primary industry will reach 9.7 billion kWh, 46.5 billion kWh, 139.7 billion kWh, and 198 billion kWh in 2030. If China continues to develop at the growth rate from 2011 to 2020, it is estimated that by 2030, the electricity usage of the tertiary industry will reach three trillion kilowatt hours, the secondary industry’s power usage will reach seven trillion kilowatt hours, and the primary industry’s electricity consumption will reach 150 billion kWh. Therefore, it is expected that the total power consumption of China's three major industries will reach 10.15 trillion kWh. When the development trend develops and changes, we can predict the scale and development trend of power consumption in various industries in advance by observing the changes of key indicators and combining the different scenarios preset in this paper. Judging from the power consumption results of China's three major industries, the power consumption of the secondary industry and the tertiary industry accounts for a large proportion, the growth rate of the power consumption of the secondary industry has slowed down, and the growth trend of the power consumption of the tertiary industry is obvious. We suggest that the government adjust the industrial structure according to the power consumption trends of different industries, build a smart grid, and formulate active and effective energy policies.

  1. Are the conclusions supported by the results?

The results support the conclusion. China's electricity demand is expected to reach 10-12 trillion kWh by 2030, according to the report "Zero Carbonizing Electricity Growth" by the Beijing Rocky Mountain Institute and the Energy Commission. According to the development trend from 2011 to 2020, the total power consumption of China's three major industries in 2030 will reach 10.15 trillion kWh. The findings of this paper are in the range of 10-12 trillion kWh in the report Zero Carbonization of Electricity Growth, so the results are supportive. At the same time, we also explored the development trend of electricity consumption under different scenarios by industry.

Thank you again for your constructive and encouraging comments.

Reviewer 2 Report

There is a need to improve the manuscript by taking the following steps:

(1) References style should be according to journal requirements. 

(2) There is also a need to highlight the significance of  Gray correlation analysis and why this study is preparing the Gray method on other gray or other econometrics-based forecasting methods.

(3) The authors should compare the current results with the previous studies. Try to highlight the contribution in terms of (data, methodology, and results). 

Author Response

Detailed Response to Reviewers

Firstly, thanks for your careful and insightful comments. The revisions are listed as follows:

Reviewer 2:

  1. References style should be according to journal requirements. 

We have reorganized the style of the references as required by the journal.

  1. There is also a need to highlight the significance of  Gray correlation analysis and why this study is preparing the Gray method on other gray or other econometrics-based forecasting methods.

When introducing the grey relational analysis model, the reasons for using the grey method are described, and the advantages and importance of the grey method are emphasized. Details as follows:

The grey correlation degree model determines the correlation order according to the correlation degree and the correlation matrix. The closer the two curves are, the greater the grey correlation degree between the corresponding sequences is, and vice versa. At the same time, compared with the traditional mathematical statistics method, the gray relational analysis model has fewer observations, and does not require too much sample size, nor does it require typical distribution laws, the calculation is simple, and the quantitative analysis results will be consistent with the qualitative analysis results. The relevant factors affecting the main factors are objectively analyzed by using the grey relational analysis method. It can explore uncertain relationships between one major factor of the system and other factors. Electricity consumption is affected by various external factors to varying degrees, which are related to factors such as population, economy, industry, and ecology. These complex factors are not completely clear in a certain period of time, we can understand it as a gray system, and use the gray correlation analysis model to explore the relationship between the electricity consumption of the three major industries and other factors in the gray system.

  1. The authors should compare the current results with the previous studies. Try to highlight the contribution in terms of (data, methodology, and results).

The main contributions of this paper can be summarized into three parts: 1) Using a grey multivariate convolution model with less error and priority of new information (GMCN(1,N)) to predict the power consumption of China's three major industries. 2) In order to accurately predict the power consumption of the three major industries, seven factors closely related to power consumption were selected for grey correlation analysis, and the key indicators affecting the power consumption of each industry were screened out. Then, forecast the impact indicators by scenario. 3) By forecasting the power consumption of the three major industries, it objectively reflects the energy and power consumption trends of the three major industries in China at different levels of development.

The comparison charts of the original data and the fitted data are shown in Figure 2, Figure 3, and Figure 4, which are easy to understand.

Figure 2. Fitting between predicted and actual values

   Figure 3. Fitting between predicted and actual values

       Figure 4. Fitting between predicted and actual values

Grey correlation analysis is carried out according to the development trend of things, and the correlation sequence is determined according to the correlation degree and the correlation matrix. The closer the curves of the two are, the greater the gray correlation degree between the corresponding sequences is. At the same time, compared with the traditional mathematical statistics method, the gray relational analysis model has fewer observations, and does not require too much sample size, nor does it require typical distribution laws, the calculation is simple, and the quantitative analysis results will be consistent with the qualitative analysis results. The relevant factors affecting the main factors are objectively analyzed by using the grey relational analysis method. It can explore uncertain relationships between one major factor of the system and other factors. In this paper, the grey multivariate model with the priority of new information is used to forecast the electricity consumption of the three major industries respectively. New information-first grey multivariate models have two advantages over traditional forecasting models. First, the model considers the priority of new information, and gives greater weight to the new information in the calculation process, which can make future predictions more accurate. Secondly, the model takes into account multiple variables, not just the forecast of a single electricity consumption. On the basis of the normal development trend, the grey multivariate model takes into account the index with the greatest correlation with electricity consumption, making the forecast more realistic . Through observation, we found that the MAPE values of the GMCN (1, 2) models of power consumption in the primary, secondary, and tertiary industries were 5.78%, 1.36%, and 2.79%, all less than 10%. The fitting effect is very good, so it can be used to predict the power consumption of the three major industries.

China's electricity demand is expected to reach 10-12 trillion kWh by 2030, according to the report "Zero Carbonizing Electricity Growth" by the Beijing Rocky Mountain Institute and the Energy Commission. According to the development trend from 2011 to 2020, the total power consumption of China's three major industries in 2030 will reach 10.15 trillion kWh. The findings of this paper are in the range of 10-12 trillion kWh in the report "Zero Carbonization of Electricity Growth", so the findings are in line with the actual situation.

Thank you again for your constructive and encouraging comments.

Reviewer 3 Report

The article is difficult to read because of many mistakes, inadequacy. Carelessness in the text submitted for review is unacceptable. It concerns mainly part 1 Method introduction

1. line 149: an inadequate citation

2. Lines 157-159: reformat this sentence/formulas

3. formula 2: You should distinguish between lowercase and uppercase letters e.g. x and X?

4. I'm afraid you are incorrectly using ksi and rho symbols in lines and ksi_{i} and gamma_{i} 164-169

5. Check the use of m  and n in formula (4)

6. Define a symbol x_{i}^(0)(1)  (formula 4)

7. Formula (6): on the left there is ^(1) but on the right there isn't. Why?

8.  Explain what does u mean (the ash consumption) and what do you assume about it. Is it stochastic?

9. Lines 205: b_{1},...b_{N} are parameters. In line 200 there was b_{n}.

10. Formula 10: Define the model you apply the least square method. Does the estimator has any properties within your assumptions?

11. Check the sentence form 288-290. I' m afraid it is incorrect or not clear.

12. Explain you explain GMCN(1,2) - why 1, why 2?

13. Line 371: I found a definition with mining included in primary industry. Check it please.

14. Could you explain why did you write lines 448-450 (I couldn't find the explanation in the text).

11. Define omega in formula (13)

12. Line 226: What do you mean "After consulting the data"? 

13. Do sensitive analysis changing ksi (=0.5) and lambda (=0,7).

14. Correct the descriptions of the tables.

15. Do you think the Russia-Ukraine war could have any impact on the forecasts?

16. It's not clear to me if you employ values from Table 5 in the forecasting procedure. If yes, could you explain why the Gray correlation might be the same (constant) in the future?

17. Why did you apply the model? You write about it something but almost at the end of the article line 432: ,,Because the GMCN(1,2) model is capable of studying nonlinear relationships and small data, it is reasonable to use the GMCN(1,2) model to analyze this set of data." Write more and earlier.

Author Response

Detailed Response to Reviewers

Firstly, thanks for your careful and insightful comments. The revisions are listed as follows:

Reviewer 3

  1. line 149: an inadequate citation.

The gray relational analysis model has a wide range of applications. We describe the reasons for using the gray method, emphasize the advantages and importance of the gray method, and add some references to show that the gray relational analysis is widely used. Specifically in Section 1 of the text.

  1. Lines 157-159: reformat this sentence/formulas.

In order to facilitate understanding, we have reorganized the method introduction of the grey relational analysis model, as shown below:

Assuming that  is a system factor and its observed data on serial number is,  is called the behavior sequence of factor .

Assuming that is the characteristic behavior sequence of the system,

  is the sequence of relevant factors.

Then the grey relation coefficient is obtained by the following formula.

The relation value can be calculated by

 is the correlation between the main behavior sequence and the related influenced factor sequence.

  1. Formula : You should distinguish between lowercase and uppercase letters e.g. x and X?

We have reorganized the method introduction of the grey relational analysis model. In the meanwhile, the upper and lower case of the letters in the formula have been adjusted. As shown below:

  1. I'm afraid you are incorrectly using and  symbols in lines and  and gamma_{i} 164-169.

Thank you for your reminder, I have rewritten the formula for the grey relational analysis model based on your suggestion. As follows:

  1. Check the use of m and n in formula (4).

Formula 4 is

m and n represent different meanings, m represents the number, and n represents the data sequence. Specific examples are as follows:

When =1, the formula is ,

When =2, the formula is ,

When =N, the formula is .

 is the first data sequence, is the second data sequence,is the Nth data sequence.

  1. Define a symbol (formula 4).

 represents the first number in the original data sequence.

  1. Formula (6): on the left there is ^(1) but on the right there isn't. Why?

Because this formula represents the first-order accumulation of the original data , the 1 on the left represents the first-order accumulation of the original data, and the 0 on the right represents the original data.

  1. Explain what does u mean (the ash consumption) and what do you assume about it. Is it stochastic?

 is not stochastic, it is an unknown parameter. By solving the least squares method, the parameter value  is obtained, which is a constant in the time response equation.

  1. Lines 205: b_{1},...b_{N} are parameters. In line 200 there was b_{n}.

We have changed  to  .

  1. Formula 10: Define the model you apply the least square method. Does the estimator has any properties within your assumptions?

Least squares finds the best functional match for the data by minimizing the sum of squared errors. The unknown data can be easily obtained by using the least squares estimation method, and the sum of squares of the errors between the obtained data and the actual data can be minimized. We assume that there are no systematic errors in the measurement system, only pure random errors.The errors are normally distributed.

Unlike traditional statistical models, grey forecasting models do not require too many assumptions. The purpose of applying the least squares method is to perform parameter fitting, find the optimal parameters, and obtain the model with the smallest error.

  1. Check the sentence form 288-290. I' m afraid it is incorrect or not clear.

Thank you for your reminder, I have revised the sentence. As follows:

The sum of the GDP of the tertiary industry and the GDP of the secondary industry exceeds 90%, occupying an absolute position, of which the tertiary industry accounts for more than 50%, becoming the largest industry.

  1. Explain you explain GMCN(1,2) - why 1, why 2?

The grey multivariable convolution model with priority accumulation of new information (GMCN(1,N)) is defined as follows: The GMCN(1,N) model is a grey prediction model with first-order differential equations and n variables. So the 1 in GMCN(1,2) means a first order differential equation and the 2 means 2 variables. Taking the GMCN(1,2) model of the electricity consumption of the tertiary industry as an example, the two variables are the electricity consumption of the tertiary industry and the GDP per capita. GDP per capita is the most influential factor related to electricity consumption in the tertiary industry.

  1. Line 371: I found a definition with mining included in primary industry. Check it please.

The primary industry mainly refers to the industries that produce food ingredients and other biological materials, including industries that directly use natural objects as production objects, such as planting, forestry, animal husbandry, and aquaculture. The secondary industry mainly refers to the processing and manufacturing industry, which uses the basic materials provided by nature and the primary industry for processing. Secondary industries include extractive industries, manufacturing, water, electricity, steam, hot water, gas and construction. The mining industry belongs to the extractive industry, which is a typical secondary industry in China.

  1. Could you explain why did you write lines 448-450 (I couldn't find the explanation in the text).

The new crown pneumonia epidemic has been going on for two years. During this period of time, the epidemic prevention and control has affected the normal production and living order, and had some negative effects on the secondary and tertiary industries. At the same time, the urgent need for materials has increased the proportion of the GDP of the primary industry, and the electricity consumption of the primary industry has also increased. Therefore, in the scenario design, we took into account the possible impact of future epidemics such as the new crown pneumonia epidemic on electricity consumption.

  1. Define in formula.

Formula :

Where

    ,

The GMCN(1,2) model consists of a first-order differential equation and two variables. The 1 in GMCN(1,2) means a first order differential equation and the 2 means 2 variables. One of the variables is the original electricity consumption series, and the other variable is the influencing factor series of electricity consumption. is the first-order cumulative sequence of influencing factors of electricity consumption. So  represents the number in the sequence of influencing factors.

  1. Line 226: What do you mean "After consulting the data"? 

By consulting relevant literature, the standards in Table 1 were obtained. Specific references have been cited in the paper.

  1. Do sensitive analysis changing (=0.5) and  (=0,7).

We selected ,,,respectively, and the result was that the gray correlation between industrial electricity consumption and economic and social indicators changed. However, after sorting the grey correlation degree between industrial electricity consumption and economic and social indicators according to the size, the correlation order of various economic and social indicators and industrial electricity consumption has not changed. It can still be found that the per capita GDP has the greatest correlation with the electricity consumption of the tertiary industry, the urbanization level has the greatest correlation with the electricity consumption of the secondary industry, and the proportion of the output value of the primary industry has the greatest correlation with its electricity consumption. So no matter what value  takes (), the final conclusion will not change.

When we obtain  ()through particle swarm optimization algorithm, the prediction model error is the smallest and the fitting effect is the best. If you change the size of the value, the error will become larger, so we don't change the size.

  1. Correct the descriptions of the tables.

We have reorganized the table and described key results.

  1. Do you think the Russia-Ukraine war could have any impact on the forecasts?

I think the Russian-Ukrainian war will have a smaller impact on the forecast. As we all know, Ukraine is a major exporter of grain. The outbreak of the war has affected grain exports, which will inevitably make China pay more attention to the primary industry, and the electricity consumption of the primary industry may also increase under the influence of policies. Of course, through the adjustment of import and export policy, the impact of the Russian-Ukrainian war on the forecast will be very small. In the scenario design, we also preset the corresponding scenario.

  1. It's not clear to me if you employ values from Table 5 in the forecasting procedure. If yes, could you explain why the Gray correlation might be the same (constant) in the future?

Compared with the traditional mathematical statistics method, the gray relational analysis model has fewer observations, and does not require too much sample size, nor does it require typical distribution laws, the calculation is simple, and the quantitative analysis results will be more consistent with the qualitative analysis results. The relevant factors affecting the main factors are objectively analyzed by using the grey relational analysis method. It can explore uncertain relationships between one major factor of the system and other factors. Electricity consumption is affected by various external factors to varying degrees, which are related to factors such as population, economy, industry, and ecology. These complex factors are not completely clear in a certain period of time, we can understand it as a gray system, and use the gray correlation analysis model to explore the relationship between the electricity consumption of the three major industries and other factors in the gray system. The grey correlation degree model determines the correlation order according to the correlation degree and the correlation matrix. The closer the two curves are, the greater the grey correlation degree between the corresponding sequences is, and vice versa. Therefore, the grey correlation model is used to calculate the correlation between different influencing factors in China's economy and society and the electricity consumption of the three major industries. The value in Table 3 is the grey correlation degree between the power consumption of each industry and each economic and social index. We can judge which index factor has the greatest correlation with the power consumption of each industry by observing the size of the grey correlation degree in Table 3. By observing Table 3, we can get per capita GDP has the greatest correlation with electricity consumption of the tertiary industry, urbanization level has the greatest correlation with electricity consumption of the secondary industry, and the proportion of primary industry output value has the greatest correlation with its electricity consumption.

  1. Why did you apply the model? You write about it something but almost at the end of the article line 432, Because the GMCN(1,2) model is capable of studying nonlinear relationships and small data, it is reasonable to use the GMCN(1,2) model to analyze this set of data." Write more and earlier.

I have added the reason for using the GMCN(1,N) model in Section 1 of the text.

In this paper, the grey multivariate model with the priority of new information is used to forecast the electricity consumption of the three major industries respectively. New information-first grey multivariate models have two advantages over traditional forecasting models. Firstly, the model considers the priority of new information, and gives greater weight to the new information in the calculation process, which can make future predictions more accurate. Secondly, the model takes into account multiple variables, not just the forecast of a single electricity consumption. On the basis of the normal development trend, the grey multivariate model takes into account the index with the greatest correlation with electricity consumption, making the forecast more realistic. Through observation, we found that the MAPE values of the GMCN(1,2) models of power consumption in the primary, secondary, and tertiary industries were 5.78%, 1.36%, and 2.79% respectively, all less than 10%. The fitting effect is very good, so it can be used to predict the power consumption of the three major industries.Therefore, the research design of this paper is suitable for solving the problem of forecasting power consumption in China's three major industries.

Thank you again for your constructive and encouraging comments.

Reviewer 4 Report

1- The authors used the Grey correlation analysis model of power consumption in three major industries. I have a question why the authors used this method. There are a lot of methods better than this one. And if there are not, why not mention the weakness of the other methods?

2- The paper has a lot of numerical results that are not important to mention and the important results are not described (17 tables have a lot of numerical results) and we can exclude some tables.

3- The authors do not mention any code for their work . We need to review and confirm the code based on the findings of this paper.

4- The style of writing introduction is not good; there is a lot of text and bad design 

5- The author does not draw the schema for their method to be easily understood by the readers.

6- The authors mention the MAP on line 223 and include table 1 for evaluation. The reference for this standard evaluating MAPE% value must be provided.

7- In line 232, the authors put a link to their data. When I click on this link, it sends me to a Chinese website and the readers do not understand Chinese, so better put the data on github or (Kaggle) to be easy to use by other researchers and describe the data.

8- the English languages must be improved 

Author Response

Detailed Response to Reviewers

Firstly, thanks for your careful and insightful comments. The revisions are listed as follows:

Reviewer 4:

  1. The authors used the Grey correlation analysis model of power consumption in three major industries. I have a question why the authors used this method. There are a lot of methods better than this one. And if there are not, why not mention the weakness of the other methods?

Grey correlation analysis is carried out according to the development trend of things, and the correlation sequence is determined according to the correlation degree and the correlation matrix. The closer the curves of the two are, the greater the gray correlation degree between the corresponding sequences is. At the same time, compared with the traditional mathematical statistics method, the gray relational analysis model has fewer observations, and does not require too much sample size, nor does it require typical distribution laws, the calculation is simple, and the quantitative analysis results will be consistent with the qualitative analysis results. The relevant factors affecting the main factors are objectively analyzed by using the grey relational analysis method. It can explore uncertain relationships between one major factor of the system and other factors. Electricity consumption is affected by various external factors to varying degrees, which are related to factors such as population, economy, industry, and ecology. These complex factors are not completely clear in a certain period of time, and we can understand them as a gray system. Therefore, the gray relational analysis model is used to explore the relationship between the electricity consumption of the three major industries and other factors in the gray system.

  1. The paper has a lot of numerical results that are not important to mention and the important results are not described (17 tables have a lot of numerical results) and we can exclude some tables.

The graphs of important results are shown in Table 6, Table 8, Table 10, Figure 3, Figure 5, and Figure 7 in the text. We have excluded some tables and re-described important results.

This paper uses the GMCN (1, N) model to predict the electricity consumption of China's three major industries. The results show that China's electricity consumption is still on the rise. Due to the uncertainty of future development, we separately discussed the electricity consumption of the three major industries under different scenarios. The factor most closely related to the power consumption of the tertiary industry is GDP per capita. When the growth rate of GDP per capita is -5%, 5%, 10% and 15% respectively, the electricity demand value of China's tertiary industry will reach 815.8 billion kWh, 2158.5 billion kWh, 3303.6 billion kWh, 4929.2 billion kWh in 2030. The relationship between the urbanization level and the electricity consumption of the secondary industry is the closest. When the growth rate of the urbanization level is -1%, 1%, 2%, and 3%, the electricity demand value of China's secondary industry in 2030 will be respectively It reached 5,737.1 billion kWh, 6,773.3 billion kWh, 6,994.5 billion kWh, and 7,229.2 billion kWh in 2030. The GDP ratio of the primary industry affects the electricity consumption of the primary industry to a large extent. When the growth rate of the GDP of the primary industry is -4%, -2%, 2%, and 4%, the electricity consumption of the primary industry in China will reach 9.7 billion kWh, 46.5 billion kWh, 139.7 billion kWh, and 198 billion kWh in 2030. When my country continues to develop at the development level from 2011 to 2020, it is expected that the total power consumption of China's three major industries will reach 10.15 trillion kWh. When the development trend develops and changes, we can predict the scale and development trend of power consumption in various industries in advance by observing the changes of key indicators and combining the different scenarios preset in this paper. Judging from the power consumption results of China's three major industries, the power consumption of the secondary industry and the tertiary industry accounts for a large proportion, the growth rate of the power consumption of the secondary industry has slowed down, and the growth trend of the power consumption of the tertiary industry is obvious. We suggest that the government adjust the industrial structure according to the power consumption trends of different industries, build a smart grid, and formulate active and effective energy policies.

  1. The authors do not mention any code for their work . We need to review and confirm the code based on the findings of this paper.

The code for GMCN(1,N) is as follows

Tlp_greyPSO.m

function z = tlp_greyPSO(in)     %Seen as an inline function of pso

nn=size(in);   %Returns the number of particle populations and the number of variables

for i=1:nn(1)

    z(i,:) = grey_model(in(i,:));

end

Main_fun.m

% To run this program, just enter the original sequence and its influence sequence

%% clear panel

close all

clc

clear

Y0 = [];  %Predicted Series: Input Fitted Data

X0 = [];  %Influencing factors: input fitted data (+ predicted data)

save main_grey.mat X0 Y0

%% PSO

addpath(genpath("PSOt"));  %add file path

range = [0,1];    %Variation range of a single parameter

range = repmat(range,1,1);    %parameter range matrix

Max_V = 0.2*(range(:,2)-range(:,1));    %The maximum speed is 20% of the variation range

Pdef = [10 100 30 2 2 0.9 0.4 40 1e-8 100 NaN 0 0];    %Generally, you can modify the first three parameters(——,the number of iterations, the number of particles)

[order_fitvalue,tr,te] = pso_Trelea_vectorized('tlp_greyPSO',1,Max_V,range,0,Pdef);    %Call the PSO core module and return the optimal order

load grey_main.mat

%% test

% MAPE = grey_model(0.7);

% load grey_main.mat

%% visualization

figure

plot(Y0);

hold on    

plot(Y0_);

Grey_model.m

% Y0/X0: Original predicted sequence/influencer% Y1/X1:累加序列

% B Y/a b:Model Least Squares Solution Construction Matrix/Solution

% Y1_:response sequence

% Y0_:forecast result

function MAPE = grey_model(W)

%% data import

load main_grey.mat

[mf,n] = size(X0); %mf is the number of fitting + predicted data; n is the number of influencing factors

[m,~] = size(Y0);  %m is the number of fitted data

%% accumulate

X1(1,:) = X0(1,:);

for j = 1:n

    for i = 2:mf

        X1(i,j) = X0(i,j) + X1(i-1,j) * W;

    end

end

Y1(1)=Y0(1);

for i = 2:m

    Y1(i) = Y0(i) + Y1(i-1) * W;

end

%% GMC

for i = 1:m-1           

    Y(i,1) = Y1(i+1) - Y1(i);    

    B(i,1) = -0.5 * (Y1(i) + Y1(i+1));   

    for j = 1:n

        B(i,j+1) = 0.5 * (X1(i,j) + X1(i+1,j));

    end

    B(i,n+2) = 1;

end

b = regress(Y,B);    %Least squares method to get the coefficients of the whitening equation

Y1_(1) = Y1(1);  

for i = 1:mf

    for j = 1:n

        bX(j) = b(j+1) * X1(i,j);   

    end

    f(i) = sum(bX) + b(n+2);    %get the f function

end

for i = 2:mf

    for j = 2:i

        F(j-1) = exp(-b(1) * (i-j+0.5)) * 0.5 *(f(j) + f(j-1));

    end

Y1_(i) = Y1(1) * exp(-b(1) * (i-1)) + sum(F);     %get time response sequence

end

%% Cumulative

Y0_(1,1) = Y1_(1);

for i = 2:mf

    Y0_(i,1) = Y1_(i) - Y1_(i-1) * W;

end

%% error test

MAPE = mean(abs(Y0 - Y0_(1:m))./Y0)*100;

save grey_main.mat Y1 X1 B Y b Y1_ Y0_

end

  1. The style of writing introduction is not good; there is a lot of text and bad design .

The introduction is more clearly expressed in Section 1of this paper.

  1. The author does not draw the schema for their method to be easily understood by the readers.

The following simple flow chart is drawn in figure 1. We have drawn and improved the fitting images and predicted images of the three major industrial GMCN (1, 2) models to make the observation more intuitive and clear, as follows:

 Figure 5. The process of three major industrial power forecasting.

Figure 6. Fitting between predicted and actual values of the tertiary industry

Figure 7. Forecast results of power consumption of China’s tertiary industry.

Figure 8. Fitting between predicted and actual values of the secondary industry.

Figure 9. Forecast results of power consumption of China's secondary industry.

     Figure 10. Fitting between predicted and actual values of the primary industry.

Figure 11. Power consumption forecast results of China’s primary industry.

  1. The authors mention the MAPEon line 223 and include table 1 for evaluation. The reference for this standard evaluating MAPE% value must be provided.

Our criteria for evaluating MAPE values are derived from Table 2 in the following article[1], and specific references have been added to the article.

[1]Wang Meng,Wang Wei,Wu Lifeng. Application of a new grey multivariate forecasting model in the forecasting of energy consumption in 7 regions of China. Energy, 2022, 243.

  1. In line 232, the authors put a link to their data. When I click on this link, it sends me to a Chinese website and the readers do not understand Chinese, so better put the data on github or (Kaggle) to be easy to use by other researchers and describe the data.

We put China's economic and social data from 2011 to 2020 on github, and the usage data can be downloaded through the link https://github.com/han616807/hello.

  1. The English languages must be improved.

Thank you for your advice. We have carefully checked and improved the English writing in the revised manuscript.

Thank you again for your constructive and encouraging comments.

Round 2

Reviewer 1 Report

The current form of the manuscript does not warrant a change in my previous remarks. The article does not meet the requirements of the scientific article. What the Authors consider their achievement is an engineering achievement, not a scientific one. Scientific achievements cannot be found in the article.

Reviewer 4 Report

The paper is improved significantly, therefore I recommend its publication in its present form